# Maternal hypertensive disorders in pregnancy and early childhood cardiometabolic risk factors: The Generation R Study

**Dionne V. Gootjes** [1,2] *, **Anke G. Posthumus** [1,2], **Vincent W. V. Jaddoe** [2,3], **Bas B. van Rijn** [1,2], **Eric A. P. Steegers** [1,2]

**1** Division of Obstetrics and Fetal Medicine, Department of Obstetrics and Gynaecology, Erasmus University Medical Centre, Rotterdam, The Netherlands, **2** Generation R Study Group, Erasmus University Medical Centre, Rotterdam, The Netherlands, **3** Department of Paediatrics, Erasmus University Medical Centre, Rotterdam, The Netherlands

* d.gootjes@erasmusmc.nl

**Data Availability Statement:** Data are available from the Generation R database (contact via datamanagementgenr@erasmusmc.nl) for

## Abstract

The objective of this study was to determine the associations between hypertensive disorders of pregnancy and early childhood cardiometabolic risk factors in the offspring. Therefore, 7794 women from the Generation Rotterdam Study were included, an ongoing population-based prospective birth cohort. Women with a hypertensive disorder of pregnancy were classified as such when they were affected by pregnancy induced hypertension, pre-eclampsia or the haemolysis, elevated liver enzymes and low platelet count (HELLP) syndrome during pregnancy. Early childhood cardiometabolic risk factors were defined as the body mass index at the age of 2, 6, 12, 36 months and 6 years. Additionally, it included systolic blood pressure, diastolic blood pressure, total fat mass, cholesterol, triglycerides, insulin and clustering of cardiometabolic risk factors at 6 years of age. Sex-specific differences in the associations between hypertensive disorders and early childhood cardiometabolic risk factors were investigated. Maternal hypertensive disorders of pregnancy were inversely associated with childhood body mass index at 12 months (confounder model: -0.15 SD, 95% CI -0.27; -0.03) and childhood triglyceride at 6 years of age (confounder model: -0.28 SD, 95% CI -0.45; -0.10). For the association with triglycerides, this was only present in girls. Maternal hypertensive disorders of pregnancy were not associated with childhood body mass index at 2, 6 and 36 months. No associations were observed between maternal hypertensive disorders of pregnancy and systolic blood pressure, diastolic blood pressure, body mass index, fat mass index and cholesterol levels at 6 years of age. Our findings do not support an independent and consistent association between maternal hypertensive disorders of pregnancy and early childhood cardiometabolic risk factors in their offspring. However, this does not rule out possible longer term effects of maternal hypertensive disorders of pregnancy on offspring cardiometabolic health.

researchers who meet the criteria for access to confidential data.

**Funding:** The Generation R Study is made possible by financial support from Erasmus Medical Center, Erasmus University Rotterdam, Rotterdam, and the Netherlands Organization for Health Research and Development (ZonMw). The funders had no role in the design of the study, the data collection and analyses, the interpretation of data, or the preparation of, review of, and decision to submit the manuscript.

**Competing interests:** The authors have declared that no competing interests exist.

## Introduction

Hypertensive disorders of pregnancy (HDP) complicate up to 10% of pregnancies and represent a significant cause of morbidity and mortality in both mother and child [1–3]. After having a HDP, there is an approximately twofold risk of developing cardiovascular or cerebrovascular disease [4–6]. In contrast, conflicting data exist on the associations with cardiometabolic risk factors in the offspring [7–11].

A number of mechanism are proposed, through which HDP may affect cardiometabolic risk factors in the offspring. First, there may be alterations in fetal vasculature and cardiac development due to exposure to maternal angiogenic factors during pregnancy [12, 13]. Second, relative fetal undernutrition due to maternal vasoconstriction may lead to adjusted fetal programming, which has a negative effect on cardiometabolic health in the offspring [14–16]. Thirdly, shared maternal and fetal genetic risk and life style factors for cardiometabolic risk factors may explain the association [17–19]. Lastly, spontaneous or iatrogenic preterm birth and the associated low birthweight may mediate the association with increased cardiometabolic risk factors in the offspring [20–22].

The association between childhood cardiometabolic risk factors and the cardiometabolic profile in adult life has been well established [23, 24]. Early identification of children at risk for the development of such an adverse profile is therefore important to potentially mitigate these risks [25]. There are no consistent results with regard to an increased cardiometabolic risk for young offspring that is prenatally exposed to HDP. Therefore, we wish to add to the evidence [13, 26–29]. Thus, the aim of this study is to investigate the associations between maternal HDP and early childhood cardiometabolic risk factors, in a large and multi-ethnic population-based cohort.

## Methods

### Population and study design

This prospective cohort study was embedded in the Generation R Study, a prospective population-based cohort in Rotterdam, the Netherlands [30]. Pregnant women were eligible for the study if they had an expected delivery date from April 2002 until January 2006 and were living in the study area in the city of Rotterdam. The following pregnancies were excluded from the analysis: twin pregnancies, terminated pregnancies, intra-uterine fetal demise and pregnancies without data on maternal hypertensive disorders or early childhood cardiometabolic risk factors (**Fig 1**). The study protocol was approved by the Medical Ethical Committee of Erasmus Medical Centre, Rotterdam (MEC 198.782/ 2001/31). Written informed consent was obtained from all participants.

### Hypertensive disorders of pregnancy

Women with a HDP were classified as such when they were affected by gestational hypertension (GH), pre-eclampsia (PE) or the haemolysis, elevated liver enzymes and low platelet count (HELLP) syndrome during pregnancy. Information on physician-diagnosed GH, PE or HELLP was retrieved from hospital charts [31]. The diagnosis was determined based on the criteria of the International Society for the Study of Hypertension in Pregnancy and according to those of the American College of Obstetricians and Gynaecologists [32]. GH was defined as a systolic blood pressure ≥140 mmHg or a diastolic blood pressure ≥90 mmHg after 20 weeks of gestation in previously normotensive women. PE was defined as de novo gestational hypertension with concurrent new onset proteinuria in a random urine sample with no evidence of urinary tract infection [32, 33]. HELLP syndrome was defined according to the class I and II

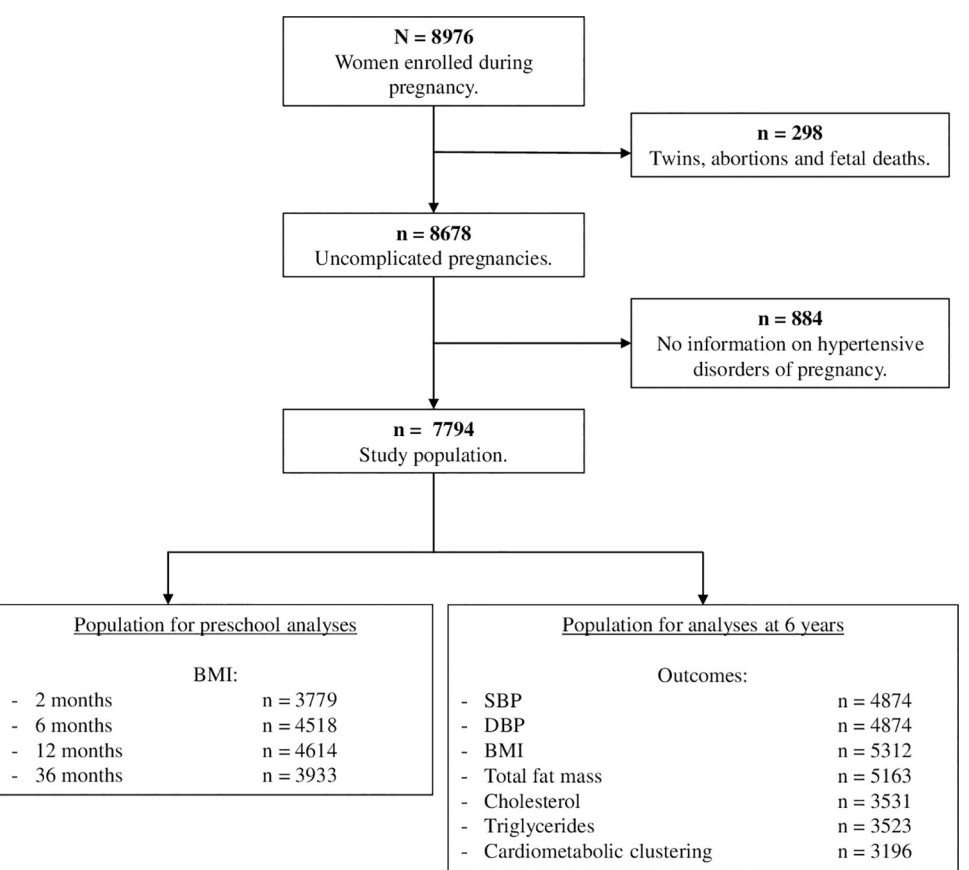

**Fig 1. Flowchart of the study population.** Abbreviations: SBP, systolic blood pressure; DBP, diastolic blood pressure; BMI, body mass index; PI, ponderal index.

2006 Mississippi criteria (platelet count $\leq$ 100 x $10^9$/L, aspartate transaminase (AST) or alanine-aminotransferase (ALT) $\geq$ 40 IU/L and lactic acid dehydrogenase (LDH) $\geq$ 600 IU/L) [34]. There were 293 cases of GH, 139 cases of PE, 14 cases of HELLP, and 45 cases which were classified as both PE and HELLP.

### Child cardiometabolic risk factors

**Body mass index and ponderal index.** Information on early childhood height and weight was collected from the community health centres, which the children visited at the age of 2 months, 6 months, 12 months and 36 months. At the age of 6 years, all children were invited to the dedicated research facility in the Erasmus University Medical Centre, Sophia Children's Hospital, for blood withdrawal and detailed measurements, among which height and weight measurements. Height and weight of children was measured according to standardized procedures: wearing underwear only, and height was measured in a barefooted standing position [30]. Body mass indexes (BMI's) were calculated as weight/height$^2$. Sex- and age- adjusted standard deviation scores (SDS) of childhood BMI were calculated, based on Dutch reference growth charts (Growth Analyzer 4.0, Dutch Growth Research Foundation) [35]. Since ponderal index might be a better measure than BMI in infancy, sensitivity analyses by using the ponderal index were performed (weight/height$^3$) [36].

**Blood pressure.** Maternal systolic and diastolic blood pressure were measured at the visit to the research facility in late pregnancy, i.e. $\geq$ 25 weeks of pregnancy. They were measured at

the right brachial artery, four times with one minute intervals, using the validated automatic sphygmanometer Datascope Accutor Plus TM (Paramus, NJ, USA) [37]. The mean value for systolic and diastolic blood pressure was calculated using the last three blood pressure measurements of each participant.

**Blood measurements.** A 30-minute fasting venous blood sample was obtained, in which total cholesterol, low-density lipoprotein cholesterol, high-density lipoprotein cholesterol, triglycerides and insulin were measured.

**Fat mass index.** Body fat was measured by Dual-Energy X-ray absorptiometry (DXA) (iDXA, General Electrics–Lunar, 2008, Madison, WI, USA), according to standard procedures [38]. Previous studies have validated DXA against computed tomography for body fat assessment [39, 40]. Android fat mass was calculated as a percentage of total fat mass [39]. In order to obtain the fat mass index uncorrelated with height, total fat mass was divided by height$^3$, as confirmed by a log-log regression analysis [41, 42].

**Cardiometabolic clustering.** Clustering of cardiometabolic risk factors was defined as such when children had three or more of the following components: android fat mass percentage at the 75th centile or above, systolic or diastolic blood pressure at the 75th centile or above; high density lipoprotein cholesterol at the 25th centile or below or triglycerides at the 75th centile or above, and an insulin levels at the 75th centile or above [43].

## Pregnancy dating

Gestational age is the most important determinant of fetal growth, so precise dating of the pregnancy is important. In accordance with clinical guidelines, if the gestational age was below 12 weeks and 5 days and the crown-rump length (CRL) measurement was smaller than 65 mm, pregnancy dating was performed using the first ultrasound measurement of the CRL. When the gestational age was older than 12 weeks and 5 days, or the biparietal diameter (BPD) was larger than 23 mm, pregnancy dating was performed using the BPD [44].

## Potential confounding variables

Covariates in the regression models were selected based on their association with both the predictor and outcome of interest. Therefore, we conducted a Directed Acyclic Graph (DAG) analysis with a consensus meeting to identify which covariates were confounders (**S1 Fig in S1 File**) [45, 46]. Consensus was achieved by the authors regarding the current structure of our regression models (DG, AP, BR, ES). The identified confounders consist of maternal BMI, ethnicity, glucose levels, educational level, smoking during pregnancy, alcohol use during pregnancy, gestational diabetes mellitus, and the child's sex.

## Covariates

Maternal age was assessed at the intake by questionnaire. Information on maternal education level, ethnicity, parity, folic acid supplementation, smoking and alcohol consumption was assessed by questionnaires during pregnancy [30]. Information on childhood sex, gestational age at birth, birth weight and length at birth was obtained from midwifery and (obstetric) medical records [44, 47]. At enrolment maternal weight (kg) and height (cm) were measured without shoes and heavy clothing after which pregnancy BMI (kg/m$^2$) was calculated. Weight measured at enrolment and pre-pregnancy weight were highly correlated (Pearson's correlation coefficient 0.95 (P-value <0.001)) [48].

Maternal glucose concentrations were measured in nonfasting blood samples which were collected at enrolment in the study. Glucose concentration (millimoles per litre) was measured with c702 module on the Cobas 8000 analyzer (Roche, Almere, the Netherlands). Information

on gestational diabetes mellitus was obtained from medical records after delivery. Gestational diabetes mellitus was diagnosed by a community midwife or an obstetrician according to Dutch midwifery and obstetric guidelines using the following criteria: either a random glucose level >11.0 mmol/l (196 mg/dL), a fasting glucose ≥7.0 mmol/l (126 mg/dL) or a fasting glucose between 6.1 mmol/l (110 mg/dL) and 6.9 mmol/l (124 mg/dL) with a subsequent abnormal glucose tolerance test [49]. In clinical practice and for this study sample, an abnormal glucose tolerance test was defined as a glucose level greater than 7.8 mmol/l (140 mg/dL) after glucose intake.

## Statistical analyses

Statistical analyses were performed using the Statistical Package of Social Sciences version 25.0 for Windows (IBM Corp., Armonk, NY, USA). A p-value < 0.05 was considered statistically significant. First, a non-response analysis was performed to compare baseline characteristics between women included and excluded from this study. Second, using Students two-tailed t-test and chi-square tests distribution of baseline characteristics and covariates within the study population were examined. Third, the associations of HDP with early childhood outcomes were examined using linear regression models: (1) a basic model including child's sex and (2) a confounder model, which was additionally adjusted for maternal and early childhood covariates selected in the DAG analysis; maternal BMI, glucose levels, educational level, ethnicity, smoking during pregnancy, alcohol use during pregnancy and gestational diabetes mellitus. Linearity was tested by assessing distribution around a diagonal line within a residual-versus-predicted-plot. Effect modifications by maternal ethnicity, child's sex, maternal smoking behaviours, maternal BMI were investigated. When significant interactions were present (p<0.1), stratified analyses were performed. Fourth, sensitivity analyses were performed. In the first analysis differences in early childhood cardiometabolic risk factors between pregnancies affected by 1) pre-eclampsia / HELLP, 2) GH or 3) 'no HDP' were tested using one-way ANOVA and Kruskal Wallis tests. To investigate the robustness of our results, sensitivity analyses defining cases as women with severe HDP (pre-eclampsia or HELLP) were performed. Lastly analysis to find differences in observed and expected values of confounders before and after imputation were conducted.

We constructed standard deviation scores (SDS) [(observed value—mean)/SD] for early childhood outcomes to enable comparison of effect estimates. The models were tested for multicollinearity using the tolerance statistic. As tolerance was >0.20 for all variables in our models, multicollinearity was unlikely. Multiple imputation procedures for confounders with missing values, were performed, creating five imputed complete datasets. These were then pooled for analyses [50]. Missing values were pre-pregnancy BMI (19.6%), glucose levels (29.6%), educational level (9.3%), ethnicity (5.7%), smoking in pregnancy (12.7%), alcohol use during pregnancy (13.9%) and gestational diabetes mellitus (2.8%).

## Results

### Characteristics of the study population

Table 1 shows maternal and child characteristics of the total study population, and within the groups of women with and without HDP. In **S1 Table in S1 File**, offspring parameters according to the type of HDP that the mother experienced are demonstrated. In our study of 7794 women, 491 women (6.3%) developed a HDP. The majority of women had a normal pre-pregnancy BMI (median 22.7 kg/m2) and were non-smokers (72.5%). When investigating differences between maternal and child characteristics between women with a HDP and without a HDP, only BMI in children at 12 months of age was statistically different (mean BMI 17.47 kg/

**Table 1. Characteristics of the study population.**

| | Study population | No HDP | HDP | p-value |
|---|---|---|---|---|
| | n = 7794 | n = 7303 | n = 491 | |
| **Maternal characteristics** | | | | |
| Maternal age at enrolment (years) | 30.2 (20.2–37.9) | 30.2 (20.2–37.8) | 30.0 (20.0–38.1) | 0.81 |
| Pre-pregnancy BMI (kg/m$^2$) | 22.7 (18.6–32.4) | 22.6 (18.6–32.4) | 22.7 (18.4–33.0) | 0.75 |
| High educational level, n (%) | 3073 (39.4%) | 2878 (39.4%) | 195 (39.7%) | 0.98 |
| Dutch and Western ethnicity, n (%) | 4479 (57.5%) | 4192 (57.4%) | 287 (58.5%) | 0.53 |
| Nulliparous, n (%) | 4260 (55.4%) | 3995 (54.7%) | 265 (54.0%) | 0.87 |
| Never smoked in pregnancy, n (%) | 5651 (72.5%) | 5278 (72.3%) | 373 (76.0%) | 0.23 |
| Never drank alcohol in pregnancy, n (%) | 3940 (50.6%) | 3685 (50.5%) | 255 (51.9%) | 0.25 |
| Glucose (mmol/l) | 4.4 (0.8) | 4.4 (0.8) | 4.4 (0.9) | 0.76 |
| Systolic blood pressure | 118.3 (12.0) | 118.2 (12.0) | 119.2 (11.8) | **0.03** |
| Diastolic blood pressure | 69.0 (9.4) | 69.0 (9.3) | 69.8 (9.9) | **0.045** |
| **Child characteristics** | | | | |
| Male sex, n (%) | 3952 (50.7%) | 3714 (50.9%) | 239 (48.7%) | 0.35 |
| Gestational age at birth (weeks) | 40.1 (36.9–42.1) | 40.1 (36.7–42.1) | 40.1 (37.1–42.0) | 0.52 |
| Preterm birth, n (%) | 441 (5.7%) | 418 (5.7%) | 23 (4.7%) | 0.34 |
| Birth weight (grams) | 3415 (561) | 3417 (564) | 3400 (530) | 0.82 |

Abbreviations: BMI, body mass index; HDP, hypertensive disorder of pregnancy. HDP included: 293 cases of GH, 139 cases of PE, 14 cases of HELLP, 20 cases of PE and HELLP and 25 cases of superponated PE/HELLP. Values are percentages for categorical variables, means (SD) for continuous variables with a normal distribution, or medians (5th, 95th percentile) for continuous variables with a skewed distribution. Confounders are imputed. Non-imputed values are presented as valid percentages. Differences in baseline characteristics were tested using Students t-test, Mann-Whitney and chi-square tests.

m2 versus mean BMI 17.25 kg/m2 p-value 0.01). Systolic and diastolic blood pressure were higher in the HDP group (119.2 vs. 118.2 and 69.8 vs. 69.0, p-value 0.03 and 0.045 respectively), though differences were small. Non-response analysis showed that women included in this study were on average slightly younger (30.2 years vs. 30.6 years, p-value 0.02) and drank less alcohol (never used alcohol 50.6% vs. 51.1%, p-value <0.001) compared to women excluded from the study. No differences were observed in pre-pregnancy BMI, educational level and ethnicity between women included and excluded from the analyses (**S2 Table in S1 File**).

### Early childhood cardiometabolic risk factors

Apart from a negative association between maternal HDP and BMI at 12 months, (confounder model: -0.15 SD, 95% CI -0.27; -0.03), no associations between maternal HDP and childhood BMI at 2, 6 or 36 months were present (**Table 2**). No differences in results were observed when we used the ponderal index as outcome measurement instead of BMI at 2, 6, 12 and 36 months (**Table 2**) [36]. Analyses with pulse as a different measure of common cardiometabolic risk factors, namely the sympatho-vagal balance, did not show different results (**Table 2**). The results did not change in sensitivity analyses with only pre-eclampsia and HELLP cases (**S3 Table in S1 File**).

At 6 years of age, no associations between maternal HDP and systolic blood pressure, diastolic blood pressure, BMI, fat mass index, cholesterol or triglyceride levels were observed. Results of interaction tests demonstrated that maternal HDP were inversely associated with triglyceride levels at 6 years of age, but only in girls (confounder model -0.28 SD, 95% CI -0.45;

**Table 2. Associations between HDP and childhood cardiometabolic risk factors.**

| | Cardiometabolic risk factor | | | Model | | | | |
| --- | --- | --- | --- | --- | --- | --- | --- | --- |
| | | | | **Basic** | | | **Confounder** | |
| | | **n** | **n** | **β (95% CI)** | **p-value** | | **β (95% CI)** | **p-value** |
| | | | **HDP** | | | | | |
| **2 months** | **BMI** | **3779** | **235 (6.2%)** | -0.06 (-0.19; 0.08) | 0.41 | | -0.05 (-0.18; 0.09) | 0.50 |
| | **PI** | **3779** | **235 (6.2%)** | -0.09 (-0.24; 0.07) | 0.28 | | -0.08 (-0.24; 0.07) | 0.31 |
| **6 months** | **BMI** | **4518** | **267 (5.9%)** | -0.11 (-0.23; 0.02) | 0.10 | | -0.09 (-0.21; 0.04) | 0.17 |
| | **PI** | **4518** | **267 (5.9%)** | -0.13 (-0.25; -0.003) | **0.045** | | -0.11 (-0.23; 0.02) | 0.09 |
| **12 months** | **BMI** | **4614** | **283 (6.1%)** | **-0.16 (-0.28; -0.04)** | **0.01** | | **-0.15 (-0.27; -0.03)** | **0.02** |
| | **PI** | **4614** | **283 (6.1%)** | **-0.17 (-0.29; -0.05)** | **0.01** | | **-0.17 (-0.29; -0.05)** | **0.01** |
| **36 months** | **BMI** | **3933** | **263 (6.7%)** | -0.03 (-0.16; 0.10) | 0.67 | | -0.002 (-0.13; 0.13) | 0.97 |
| | **PI** | **3933** | **263 (6.7%)** | -0.09 (-0.22; 0.04) | 0.18 | | -0.06 (-0.19; 0.07) | 0.35 |
| **6 years** | **BMI¶** | **5312** | **343 (6.5%)** | 0.02 (-0.08; 0.12) | 0.71 | | 0.03 (-0.07; 0.13) | 0.53 |
| | **Systolic blood pressure** | **4874** | **321 (6.6%)** | 0.04 (-0.08; 0.15) | 0.50 | | 0.05 (-0.07; 0.16) | 0.42 |
| | **Diastolic blood pressure** | **4874** | **321 (6.6%)** | 0.09 (-0.02; 0.21) | 0.10 | | 0.10 (-0.01; 0.21) | 0.09 |
| | **Fat mass index ¶** | **5163** | **330 (6.4%)** | 0.03 (-0.08; 0.13) | 0.62 | | 0.04 (-0.06; 0.14) | 0.42 |
| | **Cholesterol¶** | **3531** | **241 (6.8%)** | 0.01 (-0.12; 0.14) | 0.93 | | 0.01 (-0.12; 0.14) | 0.87 |
| | **Triglycerides¶** | **3523** | **239 (6.8%)** | -0.10 (-0.23; 0.03) | 0.13 | | -0.10 (-0.23; 0.03) | 0.14 |
| | **Pulse** | **4873** | **347 (7.1%)** | 1.10 (-0.09; 2.29) | 0.07 | | 1.18 (-0.01; 2.37) | 0.05 |
| | **Cardiometabolic risk factor clustering** | **3196** | **217 (6.8%)** | 1.15 (0.97; 1.35) | 0.41 | | 1.16 (0.84; 1.60) | 0.38 |

Abbreviations: BMI, body mass index; PI, ponderal index; HDP, hypertensive disorder of pregnancy. Values are regression coefficients (95% confidence interval) from (logistic) regression analyses that reflect the difference in childhood outcomes in SD scores, in pregnancies complicated by HDP versus pregnancies not complicated by HDP. Basic model was adjusted for child's sex. Confounder model includes maternal pre-pregnancy body mass index, educational level, ethnicity, smoking during pregnancy, alcohol use during pregnancy, maternal glucose levels and presence of gestational diabetes mellitus.
¶Variables were log transformed.

-0.10) (**Table 3**). Results of interaction test with maternal BMI were significant, however after stratification of the results, no differences were observed (**Table 4**). The values of confounders for the regression analyses before and after multiple imputation did not show relevant differences (**S4 Table in S1 File**).

**Table 3. HDP and childhood cardiometabolic risk factors at 6 years of age, split for child's sex.**

| | | | Boys (N = 3952) | | | Girls (N = 3842) | | |
| --- | --- | --- | --- | --- | --- | --- | --- | --- |
| **Cardiometabolic risk factor** | **Model** | **n** | **β (95% CI)** | **p-value** | **n** | **β (95% CI)** | **p-value** |
| **Fat mass index ¶** | **Basic** | 2568 | 0.09 (-0.06; 0.25) | 0.23 | 2594 | -0.04 (-0.18; 0.11) | 0.64 |
| | **Confounder** | 2568 | 0.12 (-0.02; 0.27) | 0.09 | 2594 | -0.03 (-0.17; 0.11) | 0.66 |
| **Cholesterol¶** | **Basic** | 1798 | 0.15 (-0.04; 0.33) | 0.12 | 1733 | -0.12 (-0.30; 0.07) | 0.20 |
| | **Confounder** | 1798 | 0.16 (-0.03; 0.34) | 0.09 | 1733 | -0.12 (-0.30; 0.07) | 0.23 |
| **Triglycerides¶** | **Basic** | 1796 | 0.09 (-0.11; 0.28) | 0.38 | 1727 | **-0.27 (-0.45; -0.09)** | **0.003** |
| | **Confounder** | 1796 | 0.09 (-0.10; 0.28) | 0.36 | 1727 | **-0.28 (-0.45; -0.10)** | **0.002** |

Abbreviations: HDP, hypertensive disorder of pregnancy. Values are (logistic) regression coefficients (95% confidence interval) that reflect the difference in early childhood outcomes in SD scores, in pregnancies complicated by HDP versus pregnancies not complicated by HDP. Basic model was adjusted for child's sex. The confounder model includes maternal pre-pregnancy body mass index, educational level, ethnicity, smoking during pregnancy, alcohol use during pregnancy, maternal glucose levels and gestational diabetes mellitus.
¶Variables were log transformed.

**Table 4. HDP and early childhood cardiometabolic risk factors at 6 years of age, split for maternal pre-pregnancy BMI.**

| Cardiometabolic risk factor | Model | <18.5 (N = 315) β (95% CI) | p-value | 18.5–25.0 (N = 5280) β (95% CI) | p-value | >25.0 (N = 2223) β (95% CI) | p-value |
|---|---|---|---|---|---|---|---|
| BMI¶ | Basic | 0.05 (-0.45; 0.54) | 0.86 | -0.05 (-0.17; 0.08) | 0.44 | 0.20 (-0.02; 0.42) | 0.08 |
| | Confounder | 0.06 (-0.41; 0.54) | 0.80 | -0.03 (-0.15; 0.09) | 0.64 | 0.20 (-0.02; 0.41) | 0.07 |

Abbreviations: HDP, hypertensive disorder of pregnancy; BMI, body mass index. Values are regression coefficients (95% confidence interval) that reflect the difference in early childhood outcomes in SD scores, in pregnancies complicated by HDP versus pregnancies not complicated by HDP. Basic model was adjusted for child's sex. Confounder model includes educational level, ethnicity, smoking during pregnancy, alcohol use during pregnancy, maternal glucose levels and gestational diabetes mellitus.

¶Variables were log transformed.

## Discussion

### Principal findings

In this study, no strong and independent associations between maternal hypertensive disorders of pregnancy and early childhood cardiometabolic risk factors were observed. A negative association between maternal HDP and offspring BMI at the age of 12 months was observed, however this was no longer present at 2 and 6 years of age.

### Results

Differences in systolic and diastolic blood pressure between the groups of women with and without a HDP were small. Moreover, mean blood pressures in the HDP group were relatively low. This could be explained by the fact that maternal blood pressure was measured in late pregnancy, i.e. ≥25 weeks of gestation. Thereby, the onset of a HDP could be (long) after the blood pressure measurement at the Generation R study research facility. Then, the blood pressure measurement in **Table 1** does not reflect blood pressure at the time of diagnosis. Second, a woman may be hospitalized due to a HDP, before she could attend the Generation R research facility: then her blood pressure measurement was missing. Lastly, the relatively low blood pressure could be due to the heterogeneity of the HDP group. Since hypertension isn't one of the criteria to diagnose 'HELLP syndrome', the women with HELLP in the HDP group do not increase the mean systolic or diastolic blood pressure.

In earlier studies, maternal HDP has been associated with a lower BMI in the offspring [7, 9]. However, data are inconsistent and associations with higher BMI have also been demonstrated [51]. Additionally, in literature, associations of PE with offspring BMI became inverse after adjustment for potential confounding factors, with maternal pre-pregnancy BMI as the main covariate attributable to this change [7]. In our analyses, inverse associations were already present in the basic analyses, before adjusting for maternal pre-pregnancy BMI. This is possibly due to the small differences in BMI between women with and without a HDP in our study population.

Our findings are in line with the results of a previous study in the same cohort as the current study. That previous study demonstrated a strong association between an adverse maternal cardiometabolic profile and an adverse cardiometabolic profile in their offspring. Moreover, they demonstrated that this association was not attenuated by pregnancy complications such as preeclampsia [52]. This endorses that the effect of PE on the offspring cardiometabolic profile is only limited.

Similar to two other studies, we found no association between maternal HDP and offspring blood pressure [53]. This may in part be explained by the challenges of obtaining a reliable

blood pressure measurement in young children. Since a physiologic childhood blood pressure has a smaller physiologic range compared to the adult blood pressure, it is harder to detect a (statistically significant) association with blood pressure in childhood. To address this point, the child's pulse was added to our outcome measures. This measure is more variable, but this did not change the results.

Next, the presence of maternal HDP was found to be inversely associated with offspring triglyceride levels, but only in girls. In literature, sex differences in the lipid profile in healthy adults have been described. It is known that since sex hormones have the ability to modulate the lipid metabolism [54–56]. Additionally, an animal study demonstrated that in mice, PE led to sex-specific metabolomic differences in the offspring: the female fetuses showed pronounced alterations in the lipid metabolism [57]. More specifically, lipid metabolite levels that were associated with triglyceride storage were lower in the female fetuses in comparison to the male fetuses, which is in line with our findings. These sex-specific differences are proposed to be due to the significantly decreased expression of lipid transporters and lipid binding proteins in the female placentas that were exposed to PE [57]. For lipids other than triglycerides, no significant differences in the offspring were observed when comparing pregnancies complicated by a HDP and pregnancies not complicated by a HDP, which is also in line with previously published studies [13, 26, 28, 29, 58].

Many studies demonstrate that the associations between maternal HDP and cardiometabolic health in the offspring are mediated by adverse birth outcomes such as preterm birth and low birth weight [59, 60]. This amplified cardiometabolic risk, attributable to fetal growth restriction and preterm birth, is not limited to childhood but is demonstrated to persist into adulthood [61, 62]. Since no significant associations between HDP and cardiometabolic risk factors in the offspring were found in our first models, no mediation analyses with adverse birth outcomes such as preterm birth and low birth weight were performed.

### Research implications

It is required to further explore the underlying mechanisms between maternal HDP and long term cardiometabolic health in the offspring. With help of metabolomics studies, the role of shared lifestyle related factors could be elucidated in the development of both hypertensive disorders and offspring cardiometabolic risk factors.

### Strengths and limitations

The main strengths of our study are the large sample size, the prospective design of the study and the standardized procedures that were used for data collection. Moreover, this study is one of few studies to assess the associations of maternal HDP with fat mass percentage and lipid levels as measures of cardiometabolic health in early childhood [63]. In contrast, previous studies examining cardiometabolic health in offspring from women with a HDP mainly focused on BMI and blood pressure [8, 64].

Some limitations of this study also need to be addressed. First, follow-up data with regard to cardiometabolic outcomes in childhood varied from 41% to 68%. Especially response rates for measures from blood sampling (e.g. cholesterol) are lower compared to BMI measures. This may have contributed to selection bias. Second, the children in this study are relatively young and therefore large differences in cardiometabolic risk factors were not to be expected. This small variation in outcome measures makes it harder to detect statistically significant associations. Third, new guidelines state that PE is diagnosed based on the presence of de novo hypertension after 20 weeks gestation accompanied by one of the following: proteinuria, acute kidney injury, liver dysfunction, neurological features, haemolysis or thrombocytopenia, or

fetal growth restriction [1, 65]. However, in our data we could only determine the presence of de novo hypertension, proteinuria and fetal growth restriction, possibly leading to misclassification of cases. To classify HDP by severity as best as possible, PE and HELLP was separated from GH [31, 66]. Fourth, even after adjusting for a large number of potential confounders, residual confounding may still be present in the observed associations. Examples of residual confounding could include lifestyle-related characteristics such as maternal (prenatal) physical activity. Finally, the majority of women in the study population were relatively young and had a low-risk profile. Moreover, in the groups of women both affected and unaffected by HDP, the mean gestational age at birth was at term. This implies relatively mild cases of HDP within this study population. As a result, the generalizability of the findings in this study is limited.

## Conclusions

In this large, prospective, population-based cohort study, no strong and persistent associations between maternal HDP and cardiometabolic risk factors in the offspring between 2 months and 6 years of age were observed. Apart from small and favourable changes in BMI and triglycerides at some of the time points, the effects of maternal HDP on child cardiometabolic risk factors seem relatively minor. This however does not rule out effects on cardiometabolic health in the offspring in later life.

## Supporting information

**S1 File.**
(DOCX)

## Acknowledgments

We gladly acknowledge the participation of all participants and the contribution of the general practitioners, hospitals, midwives, and the pharmacies in Rotterdam and all those concerned in the Generation R Study. The Generation R Study was conducted by the Erasmus Medical Centre, Rotterdam, The Netherlands, in close collaboration with the School of Law and the Faculty of Social Sciences of Erasmus University, Rotterdam, The Netherlands. Furthermore, we gratefully acknowledge Municipal Health Service, Rotterdam area; the Rotterdam Home-care Foundation; the Stichting Trombosedienst and Arts laboratorium Rijnmond, Rotterdam.

## Author Contributions

**Conceptualization:** Dionne V. Gootjes, Anke G. Posthumus, Vincent W. V. Jaddoe, Bas B. van Rijn, Eric A. P. Steegers.

**Data curation:** Dionne V. Gootjes.

**Formal analysis:** Dionne V. Gootjes.

**Funding acquisition:** Vincent W. V. Jaddoe, Eric A. P. Steegers.

**Investigation:** Dionne V. Gootjes.

**Methodology:** Dionne V. Gootjes, Anke G. Posthumus, Bas B. van Rijn.

**Project administration:** Vincent W. V. Jaddoe, Eric A. P. Steegers.

**Supervision:** Vincent W. V. Jaddoe, Bas B. van Rijn, Eric A. P. Steegers.

**Writing – original draft:** Dionne V. Gootjes.

**Writing – review & editing:** Anke G. Posthumus, Vincent W. V. Jaddoe, Bas B. van Rijn, Eric A. P. Steegers.

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
