## [Decision Letter · Decision Letter 0]

2 Jul 2021

PONE-D-21-12261

Maternal hypertensive disorders in pregnancy and early childhood cardiometabolic risk factors: the Generation R Study.

PLOS ONE

Dear Dr.Gootjes,

Thank you for submitting your manuscript to PLOS ONE. After careful consideration, we feel that it has merit but does not fully meet PLOS ONE’s publication criteria as it currently stands. Therefore, we invite you to submit a revised version of the manuscript that addresses the points raised during the review process.

We look forward to receiving your revised manuscript.

Kind regards,

Linglin Xie

Academic Editor

PLOS ONE

Journal Requirements:

2. Thank you for submitting the above manuscript to PLOS ONE. During our internal evaluation of the manuscript, we found significant text overlap between your submission and the following previously published works:

file:///C:/Users/jvostmyer/Downloads/140625_Gaillard-Romy.pdf

Please revise the manuscript to rephrase the duplicated text, cite your sources, and provide details as to how the current manuscript advances on previous work. Please note that further consideration is dependent on the submission of a manuscript that addresses these concerns about the overlap in text with published work.

3. For more information on PLOS ONE's expectations for statistical reporting, please see https://journals.plos.org/plosone/s/submission-guidelines.#loc-statistical-reporting. Please update your Methods and Results sections accordingly.”

'The Generation R Study was made possible by financial support from Erasmus MC, University

Medical Centre Rotterdam, the Netherlands; the Netherlands Organization for Health Research

and Development; the Netherlands Organization for Scientific Research; the Ministry of Health,

Welfare and Sport; and the Ministry of Youth and Families. Prof. Dr. Vincent Jaddoe received

additional grants from the Netherlands Organization for Health Research and Development

(grants 907.00303 and 916.10159, and VIDI 016.136.361) and a Consolidator Grant from the

European Research Council (ERC-2014-CoG-64916).

'The author(s) received no specific funding for this work.'

7. Please upload a copy of Figure 1, to which you refer in your text on page 27. If the figure is no longer to be included as part of the submission please remove all reference to it within the text.

Additional Editor Comments (if provided):

Reviewers' comments:

Reviewer's Responses to Questions

**Comments to the Author**

1. Is the manuscript technically sound, and do the data support the conclusions?

Reviewer #1: Partly

Reviewer #2: Yes

2. Has the statistical analysis been performed appropriately and rigorously? 

Reviewer #1: Yes

Reviewer #2: Yes

3. Have the authors made all data underlying the findings in their manuscript fully available?

Reviewer #1: No

Reviewer #2: No

4. Is the manuscript presented in an intelligible fashion and written in standard English?

Reviewer #1: Yes

Reviewer #2: No

5. Review Comments to the Author

Reviewer #1: In the methods, it was described that maternal HDPs were defined at PE, GH, and HELLP. Do you have a data chart describing the average blood pressures, liver enzymes, etc. to provide evidence for these classifications? Also, do all of the PE patients exhibit proteinuria? Some literature reports discuss PE to be defined as elevated blood pressure, with or without proteinuria. If the PE cases don't have proteinuria, what is the distinguishing factor of the PE cases from the GH cases? Also, I think it would be beneficial to show the offspring parameters according to the certain HDP that the mother experienced.

There are a couple of grammatical changes that I would also suggest. A paper reads best when it is written with a consistent point of view. So instead of saying that "we found that the presence of maternal HDP was inversely associated with offspring triglyceride levels...", it may be better to reword this type of sentence to " the presence of maternal HDP was found to be inversely associated with offspring triglyceride levels..." to keep the entire paper written from the third person point of view. Additionally, the in text references should be listed before the period.

There were also a couple of times were the data was not provided. Do you have this data to make available? If not, I wouldn't include it in the results section.

Overall, this study provides some good information, but would definitely be strengthened if the offspring were continually monitored through adulthood, as you mentioned in the discussion.

Reviewer #2: The article conducted a systematic review to evaluate the relationship between maternal hypertensive disorder during pregnancy and the offspring cardiometabolic risk. The paper observed a negative association between maternal hypertension disorder and offspring BMI at 12 month. However, some major questions are also noticed:

The symptoms of cardiometabolic disorder include high blood sugar, high blood pressure, high triglycerides, low HDL, belly fat/central adiposity. One of the parameters used to determine cardiometabolic disorder in the paper is fat mass index, which will also include peripheral. Why didn’t the author use the waist to hip ratio or waist circumference?

HELLP is one of the pregnancy disorders the author evaluated in the following studies, which is diagnosed by AST, ALT, etc. However, HELLP is not always accompanied by high blood pressure. Did the author determine that the patient with HELLP indeed had hypertension?

The author found that maternal HDP is negatively associated with offspring BMI, but not ponderal index, while the author mentioned that PI could be a better measurement for infancy. It may be necessary to include the data of PI and maternal HDP. There is a common body mass measurement for infants under 2 years old, BMI for age according to WHO. This may give a more consistent result.

In addition, there is a minor issue:

In the method section, about pregnancy dating, the author mentioned CRL measurement. However no full name was given in the previous content.

6. PLOS authors have the option to publish the peer review history of their article (what does this mean?). If published, this will include your full peer review and any attached files.

Reviewer #1: No

Reviewer #2: No

---

## [Author Response · Author response to Decision Letter 0]

1 Oct 2021

We thank the reviewers and the editorial office of PLOS ONE for the opportunity to submit major revisions. We appreciate the reviewers’ elaborate suggestions that have improved our manuscript. Our answers (in bold) are as follows

Journal Requirements:

We apologize for the inconvenience and made adjustments according to the instructions, following the templates proved on the PLOS ONE website.

2. Thank you for submitting the above manuscript to PLOS ONE. During our internal evaluation of the manuscript, we found significant text overlap between your submission and the following previously published works: file:///C:/Users/jvostmyer/Downloads/140625_Gaillard-Romy.pdf

Please revise the manuscript to rephrase the duplicated text, cite your sources, and provide details as to how the current manuscript advances on previous work. Please note that further consideration is dependent on the submission of a manuscript that addresses these concerns about the overlap in text with published work. We will carefully review your manuscript upon resubmission, so please ensure that your revision is thorough.

Unfortunately, we are not able to open the link since it’s leading to a paper saved on the C-storage. Since dr. Gaillards’ thesis is about maternal hypertensive disorders within the Generation R population, little overlap on textual parts, especially the methods section, is possible. Obviously, we would like to address this point and make textual changes.

3. For more information on PLOS ONE's expectations for statistical reporting, please see https://journals.plos.org/plosone/s/submission-guidelines.#loc-statistical-reporting. Please update your Methods and Results sections accordingly.

The Methods and Results section was adapted according to the PLOS ONE’s guidelines. As suggested, the full results of any regression analysis performed is added as a supplementary file. This includes all estimated regression coefficients, their standard error, p-values, and confidence intervals, as well as the measures of goodness of fit.

The datasets generated and analyzed during the current study are not publicly available due to individual privacy consideration. However, they are available from the data managers (Claudia J. Kruithof, c.kruithof@erasmusmc.nl or datamanagementgenr@erasmusmc.nl) and Director Generation R, Vincent Jaddoe (v.jaddoe@erasmusmc.nl) after a written agreement about the use of the data made via the Technology Transfer Office of Erasmus MC. We made adjustments accordingly in the revised cover letter.

In the new, revised version of the manuscript, we added the data on pulse as outcome and the ponderal index as outcome measurement instead of BMI at 2, 6, 12 and 36 months (Table 2).

'The Generation R Study was made possible by financial support from Erasmus MC, University

Medical Centre Rotterdam, the Netherlands; the Netherlands Organization for Health Research

and Development; the Netherlands Organization for Scientific Research; the Ministry of Health,

Welfare and Sport; and the Ministry of Youth and Families. Prof. Dr. Vincent Jaddoe received

additional grants from the Netherlands Organization for Health Research and Development

(grants 907.00303 and 916.10159, and VIDI 016.136.361) and a Consolidator Grant from the

European Research Council (ERC-2014-CoG-64916).

We note that you have provided funding information that is not currently declared in your Funding Statement. However, funding information should not appear in the Acknowledgments section or other areas of your manuscript. We will only publish funding information present in the Funding Statement section of the online submission form. Please remove any funding-related text from the manuscript and let us know how you would like to update your Funding Statement. Currently, your Funding Statement reads as follows: 'The author(s) received no specific funding for this work.'

This statement is added in the cover letter: 

Funding Statement: The Generation R Study is made possible by financial support from Erasmus Medical Center, Erasmus University Rotterdam, Rotterdam, and the Netherlands Organization for Health Research and Development (ZonMw). The funders had no role in the design of the study, the data collection and analyses, the interpretation of data, or the preparation of, review of, and decision to submit the manuscript.

7. Please upload a copy of Figure 1, to which you refer in your text on page 27. If the figure is no longer to be included as part of the submission please remove all reference to it within the text.

We apologize, this figure was uploaded separately since it was assumed this was according to the submitting guidelines. It is now included in the text again.

We now listed the supporting information with name and number, including a one-line title, at the end of your manuscript file. 

Comments to the Author

1. Is the manuscript technically sound, and do the data support the conclusions?

Reviewer #1: Partly

Reviewer #2: Yes

 2. Has the statistical analysis been performed appropriately and rigorously? 

Reviewer #1: Yes

Reviewer #2: Yes

 3. Have the authors made all data underlying the findings in their manuscript fully available?

Reviewer #1: No

Reviewer #2: No

4. Is the manuscript presented in an intelligible fashion and written in standard English?

Reviewer #1: Yes

Reviewer #2: No

 5. Review Comments to the Author

 

Reviewer #1:

In the methods, it was described that maternal HDPs were defined at PE, GH, and HELLP. Do you have a data chart describing the average blood pressures, liver enzymes, etc. to provide evidence for these classifications?

As suggested by the reviewer, systolic and diastolic blood pressure measured in the third trimester of pregnancy were added to the baseline characteristics (Table 1). Unfortunately, no data on liver enzymes was present, since the diagnosis was based on medical records. Therefore, blood samples derived at the research center, do not solely represent the blood samples at the time of diagnosis in hospital.

Also, do all of the PE patients exhibit proteinuria? Some literature reports discuss PE to be defined as elevated blood pressure, with or without proteinuria. If the PE cases don't have proteinuria, what is the distinguishing factor of the PE cases from the GH cases? 

Yes, to classify PE patients, the criteria of 2001 described by the International Society for the Study of Hypertension in Pregnancy were used. So, all of the PE patients (at least) had proteinuria. 

We address this point in the discussion, since indeed there are new guidelines state that PE is diagnosed based on the presence of de novo hypertension after 20 weeks gestation accompanied by one of the following: proteinuria, acute kidney injury, liver dysfunction, neurological features, hemolysis or thrombocytopenia, or fetal growth restriction. 

Also, I think it would be beneficial to show the offspring parameters according to the certain HDP that the mother experienced.

We added an extra table with the requested information: offspring parameters according to the certain type of HDP (S1 Table).

There are a couple of grammatical changes that I would also suggest. A paper reads best when it is written with a consistent point of view. So instead of saying that "we found that the presence of maternal HDP was inversely associated with offspring triglyceride levels...", it may be better to reword this type of sentence to " the presence of maternal HDP was found to be inversely associated with offspring triglyceride levels..." to keep the entire paper written from the third person point of view. 

We thank the reviewer for this important point, and we made changes as suggested.

Additionally, the in text references should be listed before the period.

We made adjustments accordingly, and therefore listed references before the period.

There were also a couple of times were the data was not provided. Do you have this data to make available? If not, I wouldn't include it in the results section.

Now, that data is provided within the manuscript.

Overall, this study provides some good information, but would definitely be strengthened if the offspring were continually monitored through adulthood, as you mentioned in the discussion.

Indeed, we agree with the reviewer this is one of the most important discussion points within this manuscript. 

 

Reviewer #2: The article conducted a systematic review to evaluate the relationship between maternal hypertensive disorder during pregnancy and the offspring cardiometabolic risk. The paper observed a negative association between maternal hypertension disorder and offspring BMI at 12 month. However, some major questions are also noticed:

The symptoms of cardiometabolic disorder include high blood sugar, high blood pressure, high triglycerides, low HDL, belly fat/central adiposity. One of the parameters used to determine cardiometabolic disorder in the paper is fat mass index, which will also include peripheral. Why didn’t the author use the waist to hip ratio or waist circumference?

We agree with the reviewer this measure is a good outcome measurement for cardiometabolic health. However, these measures may be imprecise and do not give any insight into the amount or differential effects of visceral and subcutaneous fat compartments.

HELLP is one of the pregnancy disorders the author evaluated in the following studies, which is diagnosed by AST, ALT, etc. However, HELLP is not always accompanied by high blood pressure. Did the author determine that the patient with HELLP indeed had hypertension?

Indeed, the author is right that HELLP does not always have to be accompanied by hypertension or proteinuria. Since the HELLP syndrome is one serious manifestation of pre-eclampsia we choose to take this disorder into account as well, despite the possibility of the absence of hypertension. Lastly, gestational hypertension and pre-eclampsia: HDP cases included 293 cases of GH, 139 cases of PE, 14 cases of HELLP, 20 cases of PE and HELLP and 25 cases of superponated PE/HELLP. Meaning, that holding onto the hypertension criterion, only 14 cases would be taken out of the analyses (since 20 had PE and HELLP, and 25 cases had superponated PE/HELLP and thus already manifested hypertension in pregnancy.)

The author found that maternal HDP is negatively associated with offspring BMI, but not ponderal index, while the author mentioned that PI could be a better measurement for infancy. It may be necessary to include the data of PI and maternal HDP.

We did not observe differences in results when we used the ponderal index as outcome measurement instead of BMI at 2, 6, 12 and 36 month. These results are now included in Table 2.

There is a common body mass measurement for infants under 2 years old, BMI for age according to WHO. This may give a more consistent result.

We thank the reviewer for this point. We are aware that both ponderal index and BMI for age are strongly correlated. Additionally, changes in PI/BMI in early infancy (until 10 years of age) are associated with greater fat-mass in later life. Moreover, we found comparable results in the analyses with both offspring BMI and the measure which is more frequently used in a clinical setting: ponderal index. 

In addition, there is a minor issue: In the method section, about pregnancy dating, the author mentioned CRL measurement. However no full name was given in the previous content.

Indeed, we apologize for this, and now added the full name into the manuscript, which is crown-rump length.

 

S1 Table. Offspring parameters according to the certain HDP that the mother experienced.

---

## [Decision Letter · Decision Letter 1]

18 Nov 2021

PONE-D-21-12261R1Maternal hypertensive disorders in pregnancy and early childhood cardiometabolic risk factors: the Generation R Study.PLOS ONE

Dear Dr. Gootjes,

Thank you for submitting your manuscript to PLOS ONE. After careful consideration, we feel that it has merit but does not fully meet PLOS ONE’s publication criteria as it currently stands. Therefore, we invite you to submit a revised version of the manuscript that addresses the points raised during the review process.

We look forward to receiving your revised manuscript.

Kind regards,

Linglin Xie

Academic Editor

PLOS ONE

Reviewers' comments:

Reviewer's Responses to Questions

**Comments to the Author**

1. If the authors have adequately addressed your comments raised in a previous round of review and you feel that this manuscript is now acceptable for publication, you may indicate that here to bypass the “Comments to the Author” section, enter your conflict of interest statement in the “Confidential to Editor” section, and submit your "Accept" recommendation.

Reviewer #1: All comments have been addressed

Reviewer #2: All comments have been addressed

2. Is the manuscript technically sound, and do the data support the conclusions?

Reviewer #1: Yes

Reviewer #2: Yes

3. Has the statistical analysis been performed appropriately and rigorously? 

Reviewer #1: Yes

Reviewer #2: N/A

4. Have the authors made all data underlying the findings in their manuscript fully available?

Reviewer #1: Yes

Reviewer #2: Yes

5. Is the manuscript presented in an intelligible fashion and written in standard English?

Reviewer #1: Yes

Reviewer #2: Yes

6. Review Comments to the Author

Reviewer #1: I found very few instances of grammatical errors, so one more round of proof reading should occur before submission. Additionally, in Table 1, which defines the maternal characteristics of the sample populations, the systolic and diastolic blood pressures are only slightly elevated. 118 vs 119 (systolic) and 69.0 vs 69.8 (diastolic) between the women without HDP vs those with HDP respectively. As mentioned in the methods, the defining characteristics of two of the three pregnancy complications are defined as systolic/diastolic blood pressures of greater than 140/80, which is relatively higher than the mean for the mothers with HDPs. Can the authors explain why the mean of the HDP was so low? Is it due to the HELLP syndrome mothers not exhibiting high blood pressure?

Reviewer #2: (No Response)

7. PLOS authors have the option to publish the peer review history of their article (what does this mean?). If published, this will include your full peer review and any attached files.

Reviewer #1: No

Reviewer #2: No

---

## [Author Response · Author response to Decision Letter 1]

21 Nov 2021

We thank the reviewers and the editorial office of PLOS ONE for the opportunity to again submit major revisions. Our answers (between **) are as follows

Reviewers' comments:

Reviewer's Responses to Questions

Comments to the Author

1. If the authors have adequately addressed your comments raised in a previous round of review and you feel that this manuscript is now acceptable for publication, you may indicate that here to bypass the “Comments to the Author” section, enter your conflict of interest statement in the “Confidential to Editor” section, and submit your "Accept" recommendation.

Reviewer #1: All comments have been addressed

Reviewer #2: All comments have been addressed

2. Is the manuscript technically sound, and do the data support the conclusions?

Reviewer #1: Yes

Reviewer #2: Yes

3. Has the statistical analysis been performed appropriately and rigorously?

Reviewer #1: Yes

Reviewer #2: N/A

4. Have the authors made all data underlying the findings in their manuscript fully available?

Reviewer #1: Yes

Reviewer #2: Yes

5. Is the manuscript presented in an intelligible fashion and written in standard English?

Reviewer #1: Yes

Reviewer #2: Yes

6. Review Comments to the Author

Reviewer #1: 

I found very few instances of grammatical errors, so one more round of proof reading should occur before submission. 

*Apologies, the manuscript was thoroughly checked again. Some mistakes have crept in after formatting the manuscript from ‘track changes’ to ‘no track changes’.*

Additionally, in Table 1, which defines the maternal characteristics of the sample populations, the systolic and diastolic blood pressures are only slightly elevated. 118 vs 119 (systolic) and 69.0 vs 69.8 (diastolic) between the women without HDP vs those with HDP respectively. As mentioned in the methods, the defining characteristics of two of the three pregnancy complications are defined as systolic/diastolic blood pressures of greater than 140/80, which is relatively higher than the mean for the mothers with HDPs. Can the authors explain why the mean of the HDP was so low? Is it due to the HELLP syndrome mothers not exhibiting high blood pressure?

*Thank you for addressing this important point. There are 3 possible explanations for the relative low (mean) blood pressures and small differences in the blood pressures between the two groups.

1. Timing of measurement.

The maternal blood pressure was measured in the third measurement period of the study, i.e. ‘late pregnancy’, which was defined as ≥25 weeks gestational age. However, hypertensive disorders of pregnancy (such as preeclampsia) are most prevalent at 34 or more weeks of gestation, with an incidence of 2.7%. von Dadelszen et al. Subclassification of preeclampsia. Hypertens Pregnancy. 2003;22(2). 

Lisonkova et al. Incidence of preeclampsia: risk factors and outcomes associated with early- versus late-onset disease. Am J Obstet Gynecol. 2013;209(6). We retrieved the diagnosis of a hypertensive disorder of pregnancy in retrospect from medical records. These blood pressure measurements therefore do not reflect the blood pressure measurement at the time of diagnosis.

2. Missing data

Before they were able to attend the measurement at the Generation R study Centre, women could already be hospitalized due to a hypertensive disorder of pregnancy.

So, these women are classified as having a HDP, and we have follow up data with regard to their infant cardiometabolic health. However, we then lack the data of (high) blood pressure in the latest phase of pregnancy. 

3. Heterogeneity of the hypertensive disorders.

Indeed, hypertension isn’t one of the criteria to diagnose ‘HELLP syndrome’, thereby the women with this diagnosis do not affect the mean blood pressure in the ‘HDP-group’.

Additionally, it is interesting to mention that the maximum blood pressures are 185 (systolic) and 118 (diastolic), indicating that indeed there are women with a hypertensive disorder of pregnancy in this group already. 

These points were added to the manuscript, chapter ‘Results’ lines 8-18:

Differences in systolic and diastolic blood pressure between the groups of women with and without a HDP were small. Moreover, mean blood pressures in the HDP group were relatively low. This could be explained by the fact that maternal blood pressure was measured in late pregnancy, i.e. ≥25 weeks of gestation. Thereby, the onset of a HDP could be (long) after the blood pressure measurement at the Generation R study research facility. Then, the blood pressure measurement in Table 1 does not reflect blood pressure at the time of diagnosis. Second, a woman may be hospitalized due to a HDP, before she could attend the Generation R research facility: then her blood pressure measurement was missing. Lastly, the relatively low blood pressure could be due to the heterogeneity of the HDP group. Since hypertension isn’t one of the criteria to diagnose ‘HELLP syndrome’, the women with HELLP in the HDP group do not increase the mean systolic or diastolic blood pressure.*

Reviewer #2: (No Response)

7. PLOS authors have the option to publish the peer review history of their article (what does this mean?). If published, this will include your full peer review and any attached files.

Do you want your identity to be public for this peer review? For information about this choice, including consent withdrawal, please see our Privacy Policy.

Reviewer #1: No

Reviewer #2: No

*PACE corrected files were uploaded.*

---

## [Decision Letter · Decision Letter 2]

1 Dec 2021

Maternal hypertensive disorders in pregnancy and early childhood cardiometabolic risk factors: the Generation R Study.

PONE-D-21-12261R2

Dear Dr. Gootjes,

We’re pleased to inform you that your manuscript has been judged scientifically suitable for publication and will be formally accepted for publication once it meets all outstanding technical requirements.

Kind regards,

Linglin Xie

Academic Editor

PLOS ONE

Additional Editor Comments (optional):

Reviewers' comments:

Reviewer's Responses to Questions

**Comments to the Author**

1. If the authors have adequately addressed your comments raised in a previous round of review and you feel that this manuscript is now acceptable for publication, you may indicate that here to bypass the “Comments to the Author” section, enter your conflict of interest statement in the “Confidential to Editor” section, and submit your "Accept" recommendation.

Reviewer #1: All comments have been addressed

2. Is the manuscript technically sound, and do the data support the conclusions?

Reviewer #1: Yes

3. Has the statistical analysis been performed appropriately and rigorously? 

Reviewer #1: Yes

4. Have the authors made all data underlying the findings in their manuscript fully available?

Reviewer #1: Yes

5. Is the manuscript presented in an intelligible fashion and written in standard English?

Reviewer #1: Yes

6. Review Comments to the Author

Reviewer #1: (No Response)

7. PLOS authors have the option to publish the peer review history of their article (what does this mean?). If published, this will include your full peer review and any attached files.

Reviewer #1: No

---

## [Editor Report · Acceptance letter]

13 Dec 2021

PONE-D-21-12261R2 

Maternal hypertensive disorders in pregnancy and early childhood cardiometabolic risk factors: the Generation R Study. 

Dear Dr. Gootjes:

I'm pleased to inform you that your manuscript has been deemed suitable for publication in PLOS ONE. Congratulations! Your manuscript is now with our production department. 

Kind regards, 

on behalf of

Dr. Linglin Xie 

Academic Editor

PLOS ONE